# Pooled analysis of oral microbiome profiles defines robust signatures associated with periodontitis

Assem Soueidan,[1,2] Katia Idiri,[2] Camille Becchina,[2] Pauline Esparbès,[2] Arnaud Legrand,[3] Quentin Le Bastard,[4,5] Emmanuel Montassier[5,6]

**ABSTRACT** Oral microbial dysbiosis has been associated with periodontitis in studies using 16S rRNA gene sequencing analysis. However, this technology is not sufficient to consistently separate the bacterial species to species level, and reproducible oral microbiome signatures are scarce. Obtaining these signatures would significantly enhance our understanding of the underlying pathophysiological processes of this condition and foster the development of improved therapeutic strategies, potentially personalized to individual patients. Here, we sequenced newly collected samples from 24 patients with periodontitis, and we collected available oral microbiome data from 24 samples in patients with periodontitis and from 214 samples in healthy individuals ($n = 262$). Data were harmonized, and we performed a pooled analysis of individual patient data. By metagenomic sequencing of the plaque microbiome, we found microbial signatures for periodontitis and defined a periodontitis-related complex, composed by the most discriminative bacteria. A simple two-factor decision tree, based on *Tannerella forsythia* and *Fretibacterium fastidiosum*, was associated with periodontitis with high accuracy (area under the curve: 0.94). Altogether, we defined robust oral microbiome signatures relevant to the pathophysiology of periodontitis that can help define promising targets for microbiome therapeutic modulation when caring for patients with periodontitis.

**IMPORTANCE** Oral microbial dysbiosis has been associated with periodontitis in studies using 16S rRNA gene sequencing analysis. However, this technology is not sufficient to consistently separate the bacterial species to species level, and reproducible oral microbiome signatures are scarce. Here, using ultra-deep metagenomic sequencing and machine learning tools, we defined a simple two-factor decision tree, based on *Tannerella forsythia* and *Fretibacterium fastidiosum*, that was highly associated with periodontitis. Altogether, we defined robust oral microbiome signatures relevant to the pathophysiology of periodontitis that can help define promising targets for microbiome therapeutic modulation when caring for patients with periodontitis.

**KEYWORDS** microbiome, periodontitis, dysbiosis, signature

Address correspondence to Emmanuel Montassier, emmanuel.montassier@univ-nantes.fr, or Assem Soueidan, assem.souiedan@univ-nantes.fr.

The authors declare no conflict of interest.

See the funding table on p. 13.

Microbes live in symbiosis with their host and play a central role in health and disease (1). Changes in the composition of the microbiome, called dysbiosis, is associated with the pathogenesis of many diseases (2). Therefore, defining microbiome biomarkers has the potential to improve disease diagnosis by offering non-invasive and personalized tools for early detection (3–5). Periodontitis affects about 30%–35% of the global population, with an increasing prevalence as people age (6). Although not fatal, periodontitis significantly reduces the quality of life and can lead to tooth loss if left untreated (7). Clinical parameters have been proposed to predict disease progression (8, 9). However, these factors were not able to adequately predict the relationship between

initial findings and prognosis. To date, reliable biomarkers are missing. Hence, development of an effective diagnosis tool is essential to identify patients at risk.

Recent studies using 16S rRNA gene sequencing analysis identified a link between microbiota composition and periodontitis. However, these studies had inherent weaknesses due to the technology used. Indeed, 16S rRNA gene sequencing does not provide sufficient resolution to differentiate multiple species (10). Sequencing just one variable region of 16S rRNA gene was not sufficient to consistently separate the bacterial species to species level. Recent research investigated the dynamic changes in the gingival microbiome within individual pockets before and after nonsurgical therapy in 12 healthy adults with chronic periodontitis. Both the taxonomic and functional compositions of the microbiome were determined using metagenomic shotgun sequencing analysis. This work, albeit limited to 12 subjects, revealed changes in the microbiota in taxonomic composition, cooccurrence of subgingival microorganisms, and functional composition that could serve as a diagnostic and prognostic indicator (7).

The features of subgingival microbiome dysbiosis associated with periodontitis have not been thoroughly compared, and this knowledge gap has limited the development of microbiome-targeted interventions for the prevention and treatment of these conditions. We hypothesized that the comparative investigation of the oral microbiome dysbiosis associated with periodontitis would enable the identification of specific signatures, thus improving the understanding of the links between this condition and paving the way to developing common and specific innovative microbiome-targeted approaches. Thus, we aimed to define robust subgingival microbiome signatures relevant to the pathophysiology of periodontitis. Here, we analyzed 262 oral microbiome samples from publicly available data sets and from a newly sequenced metagenomic data set to robustly define microbiome signatures associated with periodontitis.

## RESULTS

### A pooled metagenomic cohort of individual oral microbiome to study the subgingival microbiome in periodontitis

In order to study the role of the oral microbiome in periodontitis, we analyzed shotgun metagenomic sequencing data (11) of the gingival microbiome of 262 samples from healthy individuals and from patients with periodontitis. We profiled subjects with healthy subgingival or supragingival plaque microbiome ($n = 214$) or with periodontitis ($n = 48$). Samples were collected at the time of diagnosis by expert periodontists following a standardized protocol. For each patient, two distinct sites were sampled, the periodontitis site and the corresponding healthy contralateral site.

Quantitative taxonomic profiling of the metagenomes detected a total of 802 species present in at least one sample (average 88.4, st. dev. 46.6 per sample) and an overall functional potential of 522 microbial pathways (average 269.4, st. dev. 69.0 per sample). We observed that the bacterial fraction of the community quantitatively dominates over archaeal, micro-Eukaryotic microbes, and viral components, and we thus performed the analysis on the bacterial microbiome only.

### Microbiome diversity is altered in periodontitis compared with healthy individuals

To understand how the oral microbiome changed according to periodontitis status, we analyzed DNA isolated from gingival plaques of diseased and healthy individuals. First, we assessed the quantitative taxonomic composition of the microbiome in relation to periodontitis defined as grade B or C of the 1999 classification (12). Using this definition, we found a statistically significant difference in the microbiome composition of individuals with periodontitis compared with individuals without (permutational multivariate analysis of variance [PERMANOVA], $R^2 = 0.12$, $P = 0.001$; Fig. 1A). Consistent with the univariate analysis, we found that in periodontitis, status was the variable explaining the largest variance (periodontitis versus healthy, PERMANOVA, $R^2 = 0.11$, $P$

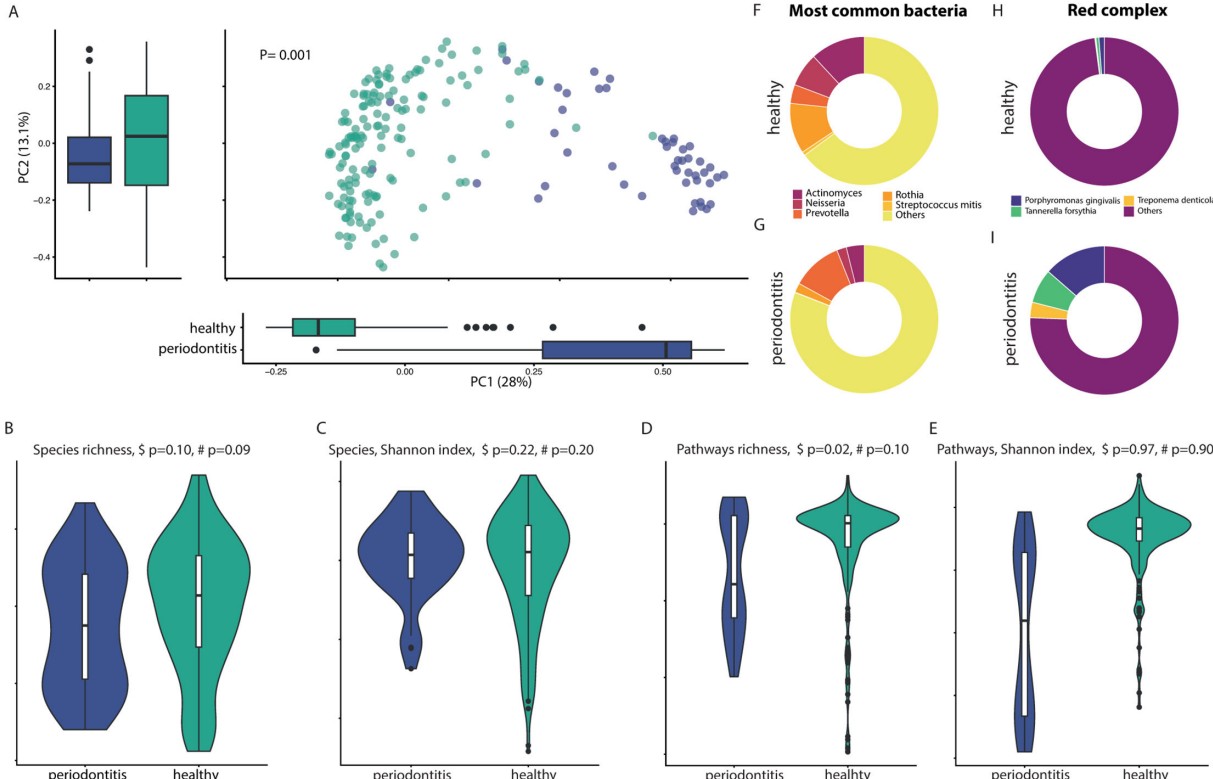

**FIG 1** (A) Principal coordinate analysis of the matrix of species distances for periodontitis sites and healthy sites. Box plots shown along each axis represent the median and interquartile range and indicate the distribution of samples along the given axis. Each point represents a single sample and is colored by site. PERMANOVA values, $R^2$ values, and $P$ values are shown. (B and C) Alpha diversity of periodontitis sites and healthy sites, depicted by the species richness and the Shannon index at species level. [#]$P$ refers to $P$ values calculated using limma linear models including study ID. [$]$P$ refers to $P$ values calculated using the Mann-Whitney U test adjusted on study ID. The lower and upper hinges of violin plots correspond to the 25th and 75th percentiles, respectively. The midline is the median. The upper and lower whiskers extend from the hinges to the largest (or smallest) value no further than ×1.5 interquartile range from the hinge, defined as the distance between the 25th and 75th percentiles. (D and E) Alpha diversity of periodontitis sites and healthy sites, depicted by the pathway richness and the Shannon index at pathway level. [#]$P$ refers to $P$ values calculated using limma linear models including study ID. [$]$P$ refers to $P$ values calculated using the Mann-Whitney U test adjusted on study ID. (F) Most common dominant bacteria in the oral microbiome in healthy individuals. (G) Most common dominant bacteria in the oral microbiome in individuals with periodontitis. (H) Triad of Socransky's red complex in healthy subjects. (I) Triad of Socransky's red complex in individuals with periodontitis.

= 0.001), and study ID was the second variable explaining the largest variance (PERMA-NOVA, $R^2 = 0.09$, $P = 0.001$). We found that principal coordinate 1 separated healthy samples from the two data sets with diseased samples (principal coordinate 1: Mann-Whitney U test adjusted on study, $P < 0.0001$; Mann-Whitney U test adjusted on study principal coordinate 2: $P = 0.90$; Fig. S1A). The overall functional potential of the microbial communities of the subgingival plaque assessed using HUMAnN3 was also distinctive for the two conditions at the level of abundance of whole-microbial pathways (PERMANOVA with multivariate analysis, periodontitis status, $R^2 = 0.37$, $P < 0.001$; study ID, $R^2 = 0.36$, $P < 0.001$; Fig. S1B). Principal coordinates 1 and 2 separated healthy samples from the two data sets with diseased samples (principal coordinate 1: Mann-Whitney U test adjusted on study, $P = 0.05$; Mann-Whitney U test adjusted on study principal coordinate 2: $P = 0.05$; Fig. S1B).

Alpha diversity was generally not associated with periodontitis status, with only pathway richness reaching statistically lower richness in periodontitis, but not Shannon index, after accounting for study as a confounding factor (Fig. 1B through E). The difference was mainly driven by one dataset [Kruskal-Wallis (KW) adjusted on study

with Dunn's test, $P < 0.001$, Fig. S1C and D for species, and $P < 0.001$, Fig. S1E and F for whole-microbial pathways].

Research demonstrated that the most common dominant bacteria in the oral microbiome were *Streptococcus mitis*, *Prevotella*, *Actinomyces*, *Neisseria*, and *Rothia*. These species constitute a core group of microbes typical of healthy oral samples (13). Here, we found that these dominant bacteria were most frequently found in the healthy samples (mean relative abundance, 35% ± 18%; range, 0%–98%) compared with the diseased samples (mean relative abundance, 19% ± 11%; range, 0%–53%; $\chi^2$, $P = 0.002$; Fig. 1F and G). The relative abundance of these dominant bacteria was not significantly different between the two data sets of diseased samples ($\chi^2$, $P = 0.70$; Fig. S2A and B).

Socransky's red complex presents a portion of the climax community in the biofilms at sites expressing progressing periodontitis, which mainly comprises species that are considered periodontal pathogens, namely, *Porphyromonas gingivalis*, *Treponema denticola*, and *Tannerella forsythia* (14–16). Here, we found that these species were rarely found in healthy samples (mean sum the relative abundance of the species of the red complex 2.1%; Fig. 1H) and were more abundant in the diseased samples (mean relative abundance of *Treponema denticola* = 3.4%, mean relative abundance of *Tannerella forsythia* = 7.4%, and mean relative abundance of *Porphyromonas gingivalis* = 13.6%; Fig. 1I). The mean relative abundance of the members of the red complex were not significantly different between the two data sets with diseased samples ($\chi^2$, $P = 0.64$; Fig. S2C and D).

## Periodontitis signatures in subgingival plaque microbiome

We pooled the samples collected in the four studies ($n = 262$ samples) and selected the top features associated with periodontitis using five different differential abundance methods. Given that "study" was the main contributor to the architecture of the gut microbiome in our data, this covariate effect was treated as a blocking factor (that is, confounder) for all subsequent analyses. We selected the top species (that is, species with different abundances between individuals with omnivore versus non-omnivore diet) found in all the five methods with a Benjamin-Hochberg (BH)-corrected $P$ value < 0.05. We identified a panel of six species that segregate periodontitis and healthy microbiomes (Fig. 2A). The ROC curve for the model based on these species was 0.93 (Fig. 2B). We also identified a panel of 15 microbial pathways with different relative abundance in periodontitis and healthy microbiomes (BH-corrected $P$ value < 0.005; Fig. 2C). The ROC curve for the model based on these pathways was 0.94 (Fig. 2D).

Next, we assessed whether and how a microbiome-based machine learning model can predict periodontitis in samples or whole cohorts not considered in the training of the classification model (17). We subsampled the combined data set of all samples ($n = 242$) in three randomly selected new data set. These random data sets contained healthy individuals and individuals with periodontitis without any overlap between the data sets. We used a Lasso-based machine learning framework to estimate the prediction ability of the combination of taxonomic and functional features of the microbiome to segregate periodontitis (cross-validation setting with nested cross-validation for feature selection) (18, 19). When exploring the relative abundance of each detected microbial species in this framework, we found substantial microbiome prediction capability, with low variability among predictions (average AUC-ROC from 0.93 to 0.97; Fig. 3A). We also obtained consistent results when the model was fit on all but one dataset and applied on the left-out one. In this setting, which tried to alleviate dataset-specific effects by considering all but one dataset in the same model, we produced prediction values averaging 0.96 and 0.98 for periodontitis (Fig. 3B through D). Functional characteristics of the microbiome profiled via a relative abundance of microbial pathways also achieved high prediction for periodontitis, with low variability and high cross-dataset consistency (average AUC-ROC from 0.92 to 0.95; Fig. 3E). We also obtained consistent results, when the model was fit on all but one dataset and applied on the left-out one. In this setting, which tried to alleviate cohort-specific effects by considering all but one dataset in the

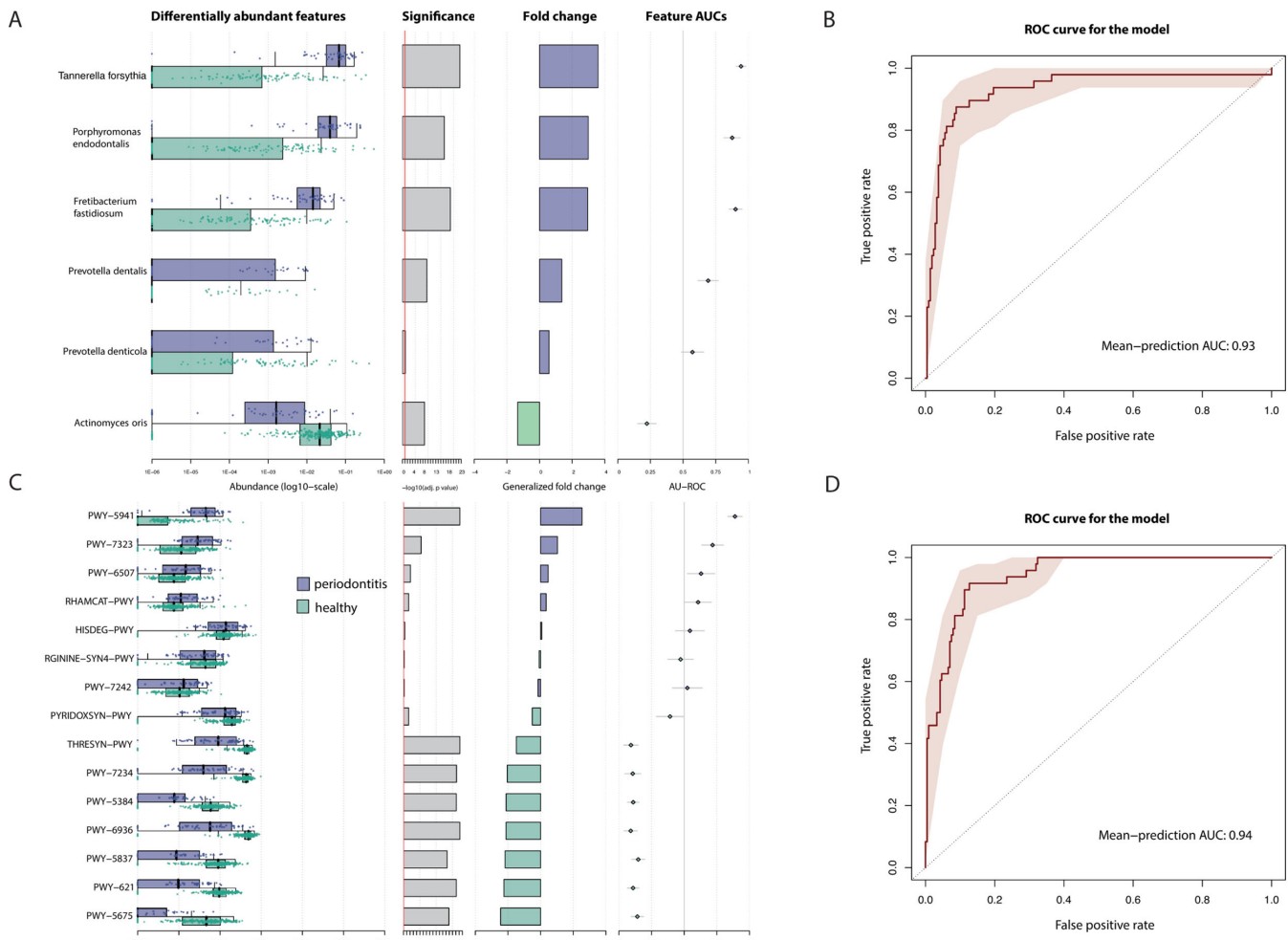

**FIG 2** (A) Significant differences at species level when comparing healthy and diseased sites using differential abundance methods and plotted with SIAMCAT. We used DESeq2 with the poscounts estimator and with apeglm, limma with TMM values, ANCOM-BC, Maaslin2, and beta-binomial regression with corncob. We keep only features that we found significant in the five methods. *P* values were corrected for multiple hypothesis testing using the Benjamin-Hochberg procedure, and a false discovery rate < 0.05 was defined as the significance threshold. (B) Machine learning association analysis between taxonomic features (species abundance) of the microbiome and periodontitis showed consistent association. Area under the curve-receiver operating characteristic (AUC-ROC) curves are computed using Lasso models trained using 100 times-repeated fivefold-stratified cross-validations. Shaded areas represent AUC-ROCs from each individual machine learning model. (C) Significant differences at pathway level when comparing healthy and diseased sites using differential abundance methods and plotted with SIAMCAT. We used DESeq2 with the poscounts estimator and with apeglm, limma with TMM values, ANCOM-BC, Maaslin2, and beta-binomial regression with corncob. We keep only features that we found significant in the five methods. *P* values were corrected for multiple hypothesis testing using the Benjamin-Hochberg procedure, and a false discovery rate < 0.005 was defined as the significance threshold. (D) Machine learning association analysis between functional profiles (pathways abundances) and periodontitis showed consistent association. AUC-ROC curves are computed using Lasso models trained using 100 times-repeated fivefold-stratified cross-validations. Shaded areas represent AUC-ROCs from each individual machine learning model.

same model, we produced prediction values averaging 0.94 and 0.97 for periodontitis (Fig. 3F through H). These taxonomic and functional predictions were not dependent on the specific machine learning approach, as adopting random forest instead of Lasso produced similar results for microbial species or functions (Fig. S3A and B, respectively), and overall, they point to substantial cross-dataset reproducible links between the microbiome and periodontitis.

We then looked for microbial taxa or functions consistently associated with periodontitis across the three datasets using MetaVolcanoR combined with differential feature expression results based on a *P* value combining approach to summarize differential feature expression from different studies. The species expression meta-analysis

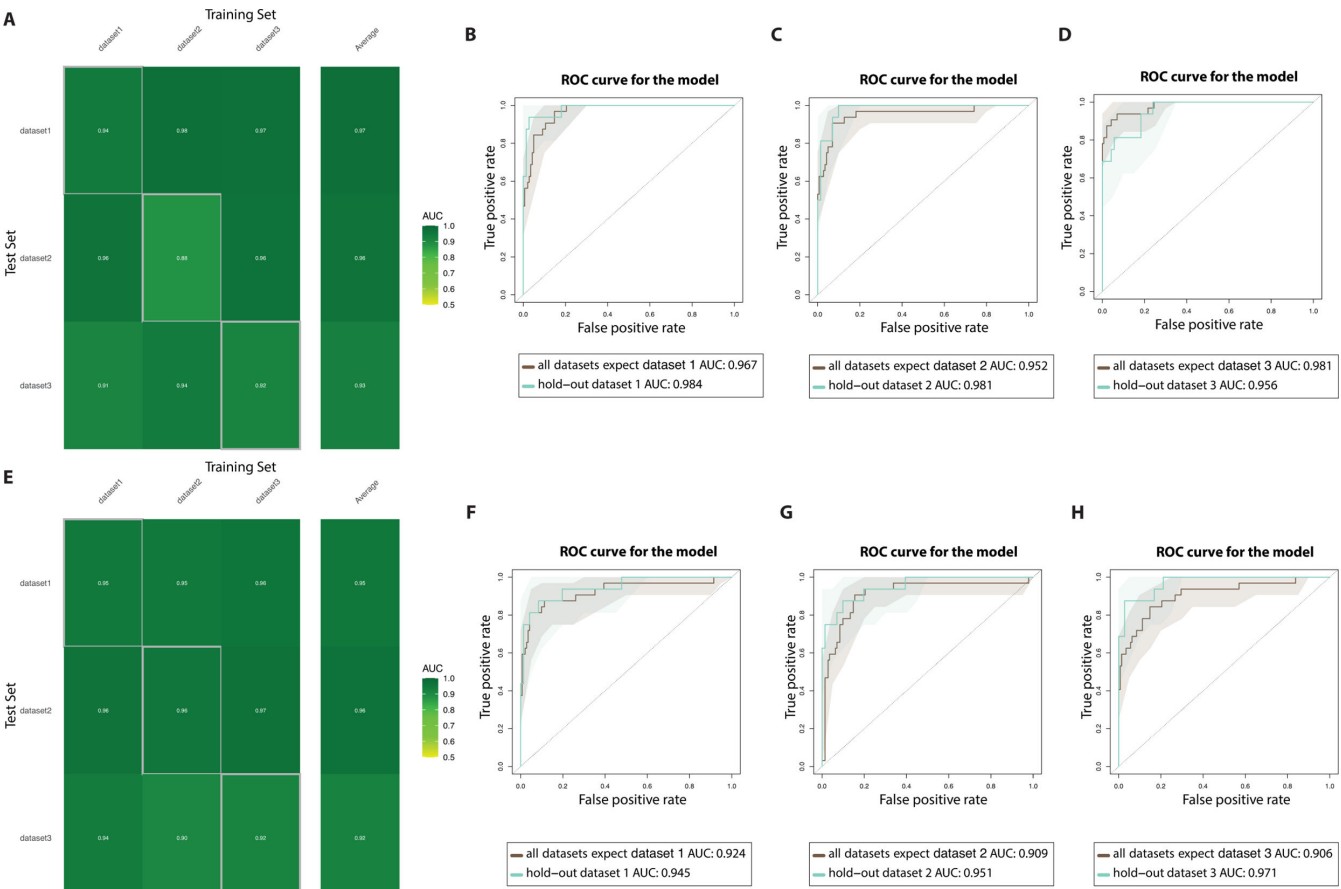

**FIG 3** (A) Prediction matrix for microbiome-based prediction of diet within each data set (values on the diagonal), across pairs of cohorts (one cohort used to train the model and the other for testing). We report the AUC-ROC values obtained from Lasso models on species-level relative abundances. Values on the diagonal refer to the median AUC-ROC values of 100 times-repeated fivefold-stratified cross-validations. Off-diagonal values refer to AUC-ROC values obtained by training the classifier on the cohort of the corresponding row and applying it to the cohort of the corresponding column. (B–D) AUC-ROC for microbiome-based prediction of periodontitis in the leave-one-dataset-out setting (training the model on all but one dataset and testing on the left-out cohort) with species. We report the AUC-ROC values obtained from Lasso models on species-level relative abundances. (E) Prediction matrix for microbiome-based prediction of diet within each data set (values on the diagonal), across pairs of cohorts (one cohort used to train the model and the other for testing). We report the AUC-ROC values obtained from Lasso models on pathway-level relative abundances. Values on the diagonal refer to the median AUC-ROC values of 100 times-repeated fivefold-stratified cross-validations. Off-diagonal values refer to AUC-ROC values obtained by training the classifier on the cohort of the corresponding row and applying it to the cohort of the corresponding column. (F–H) AUC-ROC for microbiome-based prediction of periodontitis in the leave-one-dataset-out setting (training the model on all but one dataset and testing on the left-out cohort) with pathways. We report the AUC-ROC values obtained from Lasso models on pathway-level relative abundances.

revealed that five species were associated with periodontitis: *Tannerella forsythia*, *Bulleidia extructa*, *Enterococcus casseliflavus*, Lachnospiraceae bacterium oral taxon 096, and *Campylobacter rectus* (Fig. 4A). When assessing periodontitis with the predicted functional potential of the subgingival microbiome, we identified six pathways increased in individuals with periodontitis (Fig. 4B). This included PWY-5941 (glycogen degradation II), PWY-7791 (uridine-5′-phosphate biosynthesis III), PWY-7790 (uridine-5′-phosphate biosynthesis II), PWY-7323 (GDP-mannose biosynthesis), PWY-7953 (UDP-N-acetylmuramoyl-pentapeptide biosynthesis I), and PWY-7851 (superpathway of coenzyme A biosynthesis III).

Finally, we employed a stacked machine learning model consisting of spectral clustering and FFT technique to define a parsimonious predictive score using a minimal number of criteria. FFT-based modeling offered a simple, two-factor decision tree that individually classifies samples in periodontitis versus no periodontitis based on

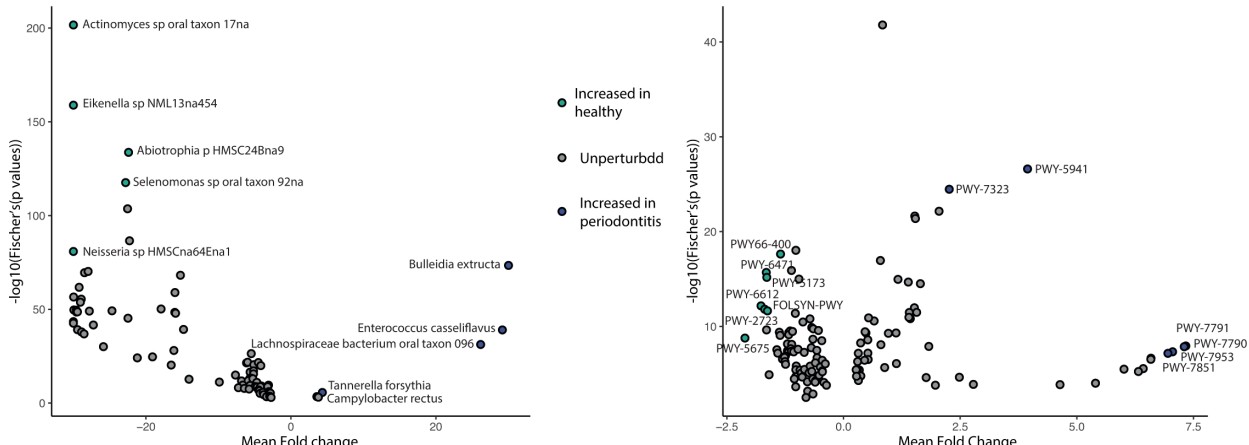

**FIG 4** (A) Random-effects model via the combining_mv function in the MetaVolcanoR R package on the random data sets on the species profiles. Random-effects *P* values obtained from each of these methods were corrected for multiple hypothesis testing using the Benjamin-Hochberg procedure. (B) Random-effects model via the combining_mv function in the MetaVolcanoR R package on the random data sets on the pathway profiles. Random-effects *P* values obtained from each of these methods were corrected for multiple hypothesis testing using the Benjamin-Hochberg procedure.

the presence of *Tannerella forsythia* and *Fretibacterium fastidiosum* (sensitivity = 93%, specificity = 84%, and area under the curve = 91%; Fig. S4A). In a test data set, the robustness of the signature was maintained (sensitivity = 95%, specificity = 100%, and area under the curve = 96%; Fig. S4B). Leveraging the signature of pathways, FFT-based modeling generated a straightforward, three-factor decision tree capable of individually classifying samples as either having periodontitis or being periodontitis free, based on the PWY66-400 (glycolysis VI), THRESYN-PWY (L-threonine biosynthesis), and the PWY-6936 (Se-amino acid biosynthesis) (sensitivity = 87%, specificity = 89%, and area under the curve = 87%; Fig. S4C). In a test data set, the robustness of the signature was maintained (sensitivity = 84%, specificity = 100%, and area under the curve = 88%; Fig. S4D).

## The periodontitis signature in other diseases

While a definitive causal relationship between periodontitis and systemic diseases is still lacking, research suggests that periodontal pathogens and the subsequent immune-inflammatory responses they trigger are linked to the development of various systemic conditions like diabetes mellitus, atherosclerotic cardiovascular diseases, and certain types of cancers (20–22). The ulcerated pocket epithelium provides a direct portal of vascular entry for periodontal pathogens, which may directly or indirectly affect other organ systems. Thus, we tested our periodontitis signature in such diseases, in two datasets for diabetes mellitus (23, 24) and in one dataset for mucositis (24). We combined our signature in a periodontitis risk index, defined as the cumulative abundance of the species associated with periodontitis minus the cumulative abundance of the species associated with the healthy state. We found that the species periodontitis signature was highly discriminative for diabetes mellitus and for mucositis (Fig. 5A, area under the curve = 0.77 [95% CI 0.62 to 0.92] and area under the curve = 0.850 [95% CI 0.79 to 0.91], respectively). The functional periodontitis signature was even better to discriminative for diabetes mellitus and for mucositis (Fig. 5B, area under the curve = 0.85 [95% CI 0.72 to 0.98] and area under the curve = 0.91 [95% CI 0.85 to 0.97], respectively).

## DISCUSSION

In our work, we investigated the microbial composition of gingival plaque in perio-dontitis, surpassing previous studies with deeper profiling (shotgun metagenomics compared with 16S rRNA sequencing), larger sample size (262 samples analyzed), and an ensemble of differential abundance methods and normalizations to assess differential

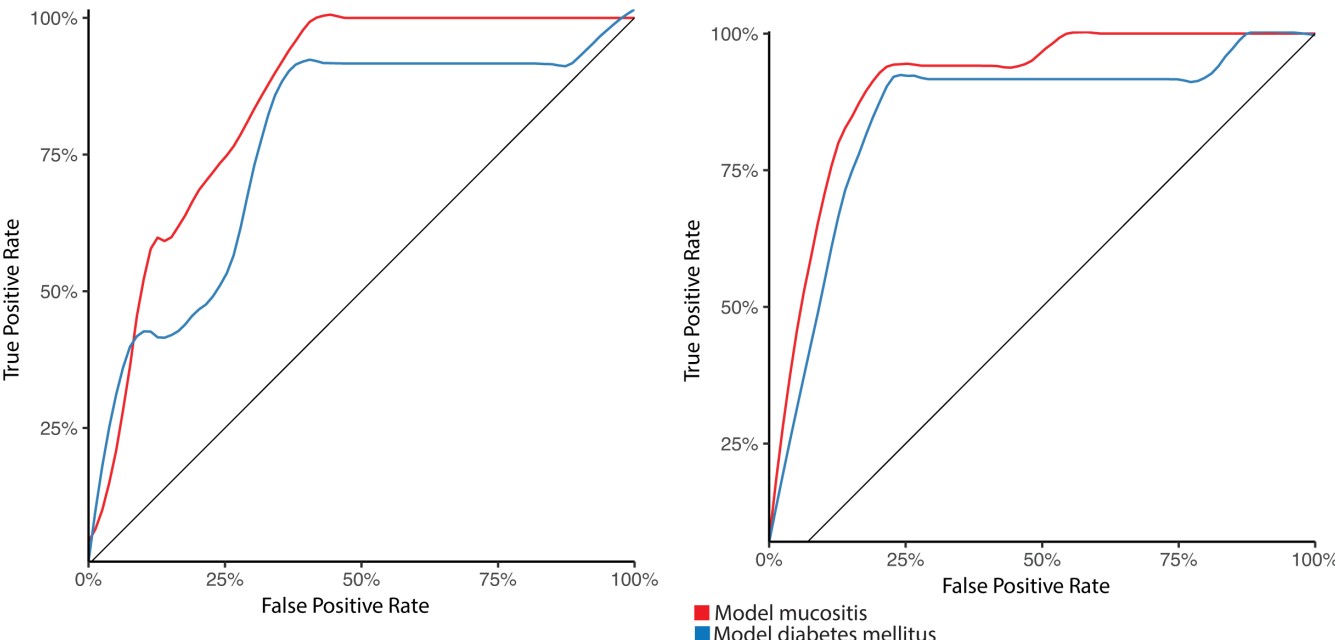

FIG 5 (A) ROC of the periodontitis risk index applied to predict type 2 diabetes and mucositis, based on species abundances. (B) ROC of the periodontitis risk index applied to predict type 2 diabetes and mucositis, based on pathway abundances.

relative abundance of species and pathways and periodontitis individuals compared with healthy subjects. We identified robust taxonomic and functional biomarkers for periodontitis and found that periodontitis exhibits a convergence toward a specific set of low-diversity pathogenic taxa. Our findings are reproducible as the link between periodontitis and the plaque microbiome is consistent across randomly selected metagenomic data sets. We define 14 bacterial species, forming a new periodontitis-related complex, that hold potential for diagnosing. Finally, a simple decision tree using only *Tannerella forsythia* and *Fretibacterium fastidiosum* accurately identifies periodontitis, suggesting they might be sufficient for diagnosis. This study's key achievement is the development and validation of frugal scores that predict individual outcomes, using a minimal set of bacteria. This observation can substantially impact the diagnosis of this condition and open doors for exploring microbiome manipulation as a promising avenue for treating this condition.

Our findings support the importance of *Fretibacterium fastidiosum* within the periodontitis-related complex, aligning with previous large-scale analyses based on 16S rRNA gene amplicon sequencing of saliva and subgingival plaque (25–27). *F. fastidiosum* that belongs to the phylum Synergistetes has been shown to be in a cluster with other periodontopathic organisms associated and shown to be elevated in abundance in periodontitis sites (28). *Fretibacterium fastidiosum* is also claimed to have co-occurrence with the red complex species, most importantly with *Tannerella forsythia* (7, 29). Szafranski et al. confirmed that *Fretibacterium fastidiosum* along with the red complex species are not only abundant but that they are also an active part in the microbial community of periodontitis (30).

We also identified *Tannerella forsythia* as a major predictor of periodontitis. This anaerobic Gram-negative bacterium resides in the subgingival niche and drives periodontal disease progression by triggering connective tissue destruction and alveolar bone resorption (31). Its arsenal of virulence factors, including BspA surface antigen (32), cell surface proteases (33), hemagglutinin (34), and cell envelope lipoproteins (35), collectively orchestrates its pathogenic potential. Moreover, the (S)-layer suppresses T-helper 17 responses in dendritic cells and exacerbates alveolar bone resorption in mice (36). Notably, polymicrobial infections involving *P. gingivalis*, *T. denticola*, and *T. forsythia*,

either as mixed or monoinfections, synergistically enhance alveolar bone resorption in rodent models (37).

We also identified that our periodontitis signature performed well to classify individuals having type 2 diabetes compared with healthy individuals. Our metagenomic study showed that several pathways implicated in glucose metabolism (PWY-5941: glycogenolysis II, PWY66-400: glycolysis VI, PWY-7323: GDP-mannose biosynthesis, RHAMCAT-PWY: L-rhamnose degradation I, PWY-7242: D-fructuronate degradation, PWY-5384: sucrose degradation IV, PWY-621: sucrose degradation I, and PWY-2723 trehalose degradation I) are altered in periodontitis, showing a potential causal relationship between periodontitis and systemic diseases. Ganesan et al. showed significantly greater levels of *Fretibacterium fastidiosum* in a diabetic cohort, suggesting a role of this species in increasing the susceptibility and progression of disease (38). We also observed two pathways linked to tetrahydrofolate (FOLSYN-PWY: superpathway of tetrahydrofolate biosynthesis and salvage; PWY-6612: superpathway of tetrahydrofolate biosynthesis), thought to contribute to the prevention of endothelial dysfunction leading to inflammatory vascular disease like atherosclerosis.

High-resolution analysis of the plaque microbiome in periodontitis unveils its crucial role in disease development. This study reveals surprisingly low variation in the overall microbial composition among individuals with periodontitis. However, we successfully identified specific bacterial species dominating these individuals, offering promise for disease stratification and classification. To refine these microbiome-based models, future studies should increase the sample size to strengthen their findings and consider key confounding factors like diabetes, smoking, alcohol consumption, and prior periodontitis history, known to influence oral health. Our results increased our understanding of the pathophysiological causes of periodontitis and could form the basis of diagnostic assays to test for clinical implementation. Our evidence synthesis can help define promising therapeutic targets for specific microbiome modulation in patients with periodontitis. However, further research is crucial to evaluate the effectiveness and safety of such personalized strategies on subgingival microbiome alterations over time, as well as the clinical implications for overall patient outcomes.

In conclusion, within the limits of our study, a signature based on *Tannerella Forsythia* and *Fretibacterium fastidiosum* described for the first time using shotgun metagenomic sequencing could allow us to refine our knowledge of the periodontal microbiome, better understand etiopathogenic mechanisms, and adapt and target therapies. A more detailed analysis of the microbiota based on the stages and grades according to the new classification of periodontal and peri-implant diseases should provide more precise data on a possible causal link between certain bacteria and the severity of the lesions. The future of research in this area should direct some efforts toward cataloging biofilm consortia to defining the major components underlying pathogenic properties. This could rely on the use of more sophisticated technologies to critically evaluate the different bacterial and immunoinflammatory signatures to better understand the etiopathogenic mechanisms of periodontitis.

## MATERIALS AND METHODS

### Metagenomic data sets included in the analysis

We collected samples from healthy patients (subgingival and supragingival plaque, $n = 125$) from the Human Microbiome Project (39), from the study of Ghensi et al. ($n = 65$) (24), and from the Shi et al. study ($n = 24$) (7), that is, 214 subgingival samples from healthy patients. We also collected samples from patients with periodontitis from the study of Shi et al. ($n = 24$).

### Newly sequenced metagenomic data set

Newly included samples were collected and processed following the protocols described in the original publication (40). Briefly, the participants were recruited from the

Department of Periodontology of the Dental Care Center of Nantes University Hospital between May 2019 and November 2019. We included 24 patients with periodontitis (2 samples per patient, a total of 48 subgingival plaque samples), diagnosed according to the clinical and radiographic criteria proposed by the 2018 International Workshop for the Classification of Periodontal Diseases and Conditions (41, 42).

The inclusion criteria involved smoking and non-smoking patients who speak and understand French; who were over 18 years old; with good general health and at least 12 teeth on the arch (excluding third molars); who presented a generalized periodontitis stage III/IV, grade B/C according to the new classification (42); who required non-surgical periodontal treatment; and who provided oral and written consent. The exclusion criteria excluded participants with acute oral lesions, endo-periodontal lesions, ulcero-necrotic gingivitis or periodontitis, and a chronic or systemic pathology or treatment that may influence the periodontal microbiota and the response to treatment (immunotherapy, corticotherapy, biotherapy, unbalanced diabetes—glycated hemoglobin [HbA1c ]≥ 7%, acute inflammatory rheumatism, and neurological deficiency); subjects using a systemic or local antibiotics in the last 3 months; and pregnant and lactating women. Subgingival plaque samples were collected as previously described (39).

The subgingival plaque samples collected using sterile paper points were kept frozen at −80°C until they were processed. Samples were then immersed in 1.5 mL phosphate-buffered saline (PBS) inside a sterile microcentrifuge tube and vortexed to dislodge the bacteria. After removing the paper points with sterile cotton pliers, the samples were centrifuged at 20,000 rpm for 5 min to pellet the bacteria at room temperature, as previously described (43). The DNA was then extracted with the QIAamp DNA Mini Kit (Qiagen Inc., Venlo, The Netherlands) following the manufacturer's protocol. Sequencing libraries were prepared using the Nextera DNA Flex Library Preparation Kit (Illumina), following the manufacturer's guidelines. Sequencing was performed on the Illumina NovaSeq 6000 platform following the manufacturer's protocols.

## Metagenome pre-processing and quality control

Newly sequenced samples were pre-processed as follows. Shortly, metagenomic reads were quality controlled, removing reads of low quality (quality score < Q20), fragmented short reads (<75 bp), and reads with >2 ambiguous nucleotides. Contaminant and host DNA was identified with Bowtie2 (v2.3.5.1) in "end-to-end" global mode, allowing confident removal of human-associated reads (hg19) (44). Remaining high-quality reads were sorted and split to create standard forward, reverse, and unpaired read output files for each metagenome.

## Profiling of metagenomic samples

A total of 48 subgingival samples from patients with periodontitis and 214 subgingival samples from healthy patients were used for metagenomic shotgun analysis. Species-level profiling was performed on all the samples with MetaPhlAn 3 with default parameters (45). Gene family and pathway profiling was performed on all the samples with HuMaAn 3 with default parameters (46).

## Statistical analysis

Statistical analyses and graphical representations were performed in R using packages vegan (version 2.5-7) (47), ggplot2 (v3.3.3) (48), and siamcat (v1.12.0) (49). Correction for multiple testing with the false discovery rate procedure was applied when appropriate. All tests were two sided except where specified otherwise. The association between metadata variables and distance matrices was assessed by PERMANOVA with the adonis function in vegan. Differences between two groups were assessed with Wilcoxon rank-sum tests. For more than two groups, the Kruskal-Wallis test with post hoc Dunn tests was used. These tests were adjusted to take into account for study effect using the coin R package. We also developed a periodontitis risk index corresponding

to the difference in a patient's total relative abundance of species associated with or without development of periodontitis as previously reported (50). In detail, this risk index was defined as the cumulative abundance of the species (or pathways) associated with periodontitis minus the cumulative abundance of the species (or pathways) associated with the healthy state. We included in the signature all the species and all the pathways found significantly associated with periodontitis or healthy state based on the methods described below.

## Meta-analysis

First, we performed an analysis of the whole cohort, that is combining all the data sets together. An ensemble of differential abundance methods and normalizations (five in total) was used to assess differential relative abundance of species and pathways between healthy individuals and individuals with periodontitis. We applied the following methods:

i.    DESeq2 (v.1.30.0) (51) with the poscounts estimator (DESeq2_poscounts) and with apeglm (52).
ii.   limma (v3.46.0) (53) with TMM values (limma_voom_TMM) (the limma package includes a voom function that transforms previously normalized counts to log counts per million, estimates a mean-variance relationship, and uses this to compute appropriate observational-level weights).
iii.  ANCOM-BC (v.1.0.1) (54), which uses a linear regression framework in log scale and accounts for sampling fraction by introducing a sample-specific bias correction that is estimated from the observed data (we used the same parameters as described in the univariate/multivariate analysis).
iv.   Maaslin2 (v.1.4.0) (55), where logit-transformed relative abundances were normalized with total-sum scaling and supplied to the maaslin2 function using the variable of interest as a fixed effect.
v.    beta-binomial regression with corncob (56) that models differential abundance and differential variability.

We applied these methods on taxonomy at species level, on functional profiles represented by pathways. Each method was adjusted on study ID to take into account for study effect. We then keep only features that we found significant in the five methods. $P$ values were corrected for multiple hypothesis testing using the Benjamin-Hochberg procedure. We then compiled the species, pathways, and ARG that were found in all statistical methods and plotted the ROC curve using the SIAMCAT R package (v.1.6.0) (49).

To aid clinical decision-making, we fitted a fast and frugal tree (FFT) to predict risk groups accurately (57). FFTs are simpler versions of decision trees and have been shown to perform competitively with random forests and to facilitate biologic interrogation by decreasing the number of key features. Here, we split the whole data set ($n = 262$) in a train set, containing 80% of the samples, and a test set, containing 20% of the samples. We used the main FFTrees() function to create FFTs for the train data set and evaluate their predictive performance on the test data set.

## Machine learning analysis

We then subsampled the combined data set of all studies ($n = 4$ studies, number of samples = 262) in three randomly selected new data set. These random data sets contained non-overlapping samples from individuals with periodontitis and healthy individuals. An ensemble of differential abundance methods and normalizations (five in total as described above) were used to estimate fold changes with their respective confidence intervals between periodontitis and healthy samples in these three datasets and supplied to a random-effects model via the combining_mv function in the MetaVolcanoR R package (v.1.4.0) (58).

Data preprocessing, model building, and model evaluation were performed using the SIAMCAT R package (v.1.6.0) (49). Species relative abundances were filtered to remove markers with low overall abundance ($1 \times 10^{-4}$ maximum abundance cutoff), log10-transformed (after adding a pseudocount of $1 \times 10^{-5}$ to avoid nonfinite values), and standardized as z-scores. Functional profiles (i.e., pathways) were preprocessed similarly but using $1 \times 10^{-6}$ as the maximum abundance cutoff and $1 \times 10^{-9}$ as a pseudocount during log transformation.

## Cross-validation

A nested cross-validation procedure was applied to calculate within-cohort accuracy by splitting data into training and test sets for 100 times-repeated, fivefold-stratified cross-validation (balancing class proportions across folds). For each split, an L1-regularized (Lasso) logistic regression model was trained on the training set, which was then used to predict the test set. The lambda parameter was selected for each model to maximize the AUC-ROC under the constraint that the model contained at least five nonzero coefficients.

## Cross-study validation

Metagenomic classifiers were trained on a single cohort, and their performance was externally assessed on all other cohorts, which were normalized for comparability in the same way as the training data set. All 500 models derived from the cross-validation on the training data set (100 times-repeated fivefold cross-validation) were applied to the hold-out data set, and median predictions were taken from all models.

## Leave one data set out

Data from one cohort were set aside as an external validation set, whereas data from the remaining cohorts were pooled as a single training set on which we implemented the same procedure as above for 100 times-repeated fivefold-stratified cross-validation. ROC were built using *P* ROC (59).

## ACKNOWLEDGMENTS

We thank the members of the MiHAR laboratory for insightful discussions, all the clinicians and the personnel of the dental office of the CHU Nantes involved in the project, and all the volunteers enrolled in the study. We thank the University of Minnesota Genomic Center for performing the metagenomic sequencing. We thank the Genomics and Bioinformatics Core Facility of Nantes (GenoBiRD, Biogenouest) for its technical support and the biological resource center for biobanking of CHU Nantes (Hôtel Dieu, Centre de Ressources Biologiques, Nantes, France, BRIF: BB-0033-00040) for its technical support.

This work was supported by the Appel d'Offre Interne CHU Nantes, Delegation for Clinical Research and Innovation under number RC 21_0250: Recherche de biomarqueurs microbiotiques indexés à la réponse pro-inflammatoire et à la sévérité clinique dans les parodontites-Etude Pilote PAROMIP PAROdontites & MIcrobiote Parodontal.

E.M., A.S., and A.L. conceived and planned the study. A.S., P.E., K.I., and C.B. organized and supervised the sampling. A.S. and P.E. collected the samples. E.M. and Q.L.B. performed the bioinformatic analysis. E.M., Q.L.B., and A.S. interpreted the analyses. E.M. and A.S. wrote the manuscript. Q.L.B., K.I., C.B., and A.L. revised the manuscript. All authors read and approved the final manuscript.

## AUTHOR AFFILIATIONS

[1]Nantes Université, CHU Nantes, INSERM, Regenerative Medicine and Skeleton, Nantes, France
[2]Department of Periodontology, Faculty of Dental Surgery, Nantes, France

³CHU Nantes, Direction de la Recherche Clinique, Nantes, France
⁴Cibles et médicaments des infections et de l'immunité, IICiMed, Nantes Université, Nantes, France
⁵CHU Nantes, Service des urgences, Nantes, France
⁶Nantes Université, Inserm, CHU Nantes, Center for Research in Transplantation and Translational Immunology, Nantes, France

## AUTHOR ORCIDs

Assem Soueidan http://orcid.org/0000-0001-8759-2165
Emmanuel Montassier http://orcid.org/0000-0002-2313-1172

## FUNDING

| Funder | Grant(s) | Author(s) |
| --- | --- | --- |
| Centre Hospitalier Universitaire de Nantes (CHU de Nantes) | RC 21_0250 | Assem Soueidan |

## AUTHOR CONTRIBUTIONS

Emmanuel Montassier, Conceptualization, Data curation, Methodology, Software, Validation, Writing – original draft.

## DATA AVAILABILITY

All metagenomes have been deposited and are available at the NCBI Sequence Read Archive under accession BioProject PRJNA744078. The underlying code for this study is publicly available and is reported in supplemental methods.

## ADDITIONAL FILES

The following material is available online.

### Supplemental Material

**Figure S1 (mSystems00930-24-S0001.pdf).** Principal-coordinate analyses and alpha diversity determinations.
**Figure S2 (mSystems00930-24-S0002.pdf).** Most common dominant bacteria in the oral microbiome and triad of the Socransky's red complex.
**Figure S3 (mSystems00930-24-S0003.pdf).** Prediction matrix.
**Figure S4 (mSystems00930-24-S0004.pdf).** Fast-and-frugal tree-based staging schemes to predict periodontitis at species level.
**Supplemental methods (mSystems00930-24-S0005.docx).** R custom codes used.
**Legends (mSystems00930-24-S0006.docx).** Legends for supplemental figures.

### Open Peer Review

**PEER REVIEW HISTORY (review-history.pdf).** An accounting of the reviewer comments and feedback.

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
