## [Reviewer comments · mSystems]

Pooled analysis of oral microbiome profiles defines robust signatures associated with periodontitis

Emmanuel Montassier, Assem Soueidan, Katia Idiri, Camille Becchina, Pauline Esparbès, Arnaud Legrand, and Quentin Le Bastard

Corresponding Author(s): Emmanuel Montassier, Université de Nantes

Review Timeline:

Submission Date:

July 12, 2024

Accepted:

July 23, 2024

Editor: Shi Huang

Reviewer(s): The reviewers have opted to remain anonymous.

Transaction Report:

DOI: <https://doi.org/10.1128/msystems.00930-24>

Re: mSystems00930-24 (Pooled analysis of oral microbiome profiles defines robust signatures associated with periodontitis)

Dear Dr. Emmanuel Montassier:

All comments have been addressed in this round of review. But some figures are not optimal for publication. Texts (AUROC) in Figure 3A and 3E are now quite small and should be enlarged.

Your manuscript has been accepted, and I am forwarding it to the ASM production staff for publication. Your paper will first be checked to make sure all elements meet the technical requirements. ASM staff will contact you if anything needs to be revised before copyediting and production can begin. Otherwise, you will be notified when your proofs are ready to be viewed.

Sincerely,
Shi Huang
Editor
mSystems